# Utilizing the Gastrointestinal Microbiota to Modulate Cattle Health through the Microbiome-Gut-Organ Axes

**DOI:** 10.3390/microorganisms10071391

**Published:** 2022-07-10

**Authors:** Christina B. Welch, Valerie E. Ryman, T. Dean Pringle, Jeferson M. Lourenco

**Affiliations:** Department of Animal and Dairy Science, University of Georgia, Athens, GA 30602, USA; vryman@uga.edu (V.E.R.); dpringle@uga.edu (T.D.P.); jefao@uga.edu (J.M.L.)

**Keywords:** ruminant, microbiome-gut-organ axes, microbiome-gut-lung axis, microbiome-gut-brain axis, microbiome-gut-mammary axis, microbiome-gut-reproductive axis

## Abstract

The microorganisms inhabiting the gastrointestinal tract (GIT) of ruminants have a mutualistic relationship with the host that influences the efficiency and health of the ruminants. The GIT microbiota interacts with the host immune system to influence not only the GIT, but other organs in the body as well. The objective of this review is to highlight the importance of the role the gastrointestinal microbiota plays in modulating the health of a host through communication with different organs in the body through the microbiome-gut-organ axes. Among other things, the GIT microbiota produces metabolites for the host and prevents the colonization of pathogens. In order to prevent dysbiosis of the GIT microbiota, gut microbial therapies can be utilized to re-introduce beneficial bacteria and regain homeostasis within the rumen environment and promote gastrointestinal health. Additionally, controlling GIT dysbiosis can aid the immune system in preventing disfunction in other organ systems in the body through the microbiome-gut-brain axis, the microbiome-gut-lung axis, the microbiome-gut-mammary axis, and the microbiome-gut-reproductive axis.

## 1. Introduction

The ruminant animal has a unique set of evolutionary advantages that allow them to digest and utilize plant biomass that is undigestible by monogastric animals [1]. These advantages come from the rumen, which acts as a fermentation chamber to degrade feedstuffs and absorb organic acids before they reach the glandular stomach [2,3]. In order to degrade feedstuffs, the rumen is inhabited by a variety of microorganisms (i.e., bacteria, protozoa, fungi, and archaea) that degrade the plant materials and produce metabolites the host animal can utilize for both maintenance and growth [2,3,4]. The rumen accounts for an estimated 80% of the total volume of the gastrointestinal tract (GIT) of the ruminant and is the site in which more than 85% of the short chain fatty acid (SCFA) production in the GIT occurs [5]. In addition, approximately 50% of a ruminant’s protein requirement is produced from GIT microbial fermentation in the form of microbial protein [6,7].

The relationship between microbes colonizing the GIT can be commensal, mutualistic, competitive, and/or predatory [8,9]. Many bacteria produce bacteriocins (i.e., an antimicrobial peptide) that target other bacteria inhabiting a similar niche, providing a competitive advantage to the bacteriocin-producing bacteria [3]. However, these microbes can coexist and benefit from each other by utilizing the end products produced by microbes occupying a different niche in a process known as cross-feeding [10,11]. The microbe–microbe interactions enhance their collective ability to maximally degrade feedstuffs and provide energy and protein to the host animal.

The symbiotic relationship existing between microorganisms colonizing the GIT and the host is mainly mutualistic in nature [12]. The metabolites produced by this microbiota play an important role in host physiology by maintaining homeostasis with the host’s immune system [13]. Due to its influence on the immune system, the GIT is the site of the majority of pathogenic entrance into the host. In order to promote animal health, the GIT microbiota can be modulated to improve the functioning of the immune system and maintain homeostasis within the host’s body [14].

The microbes’ influence is not limited to the scope of the GIT but expands to all areas and organ systems of the body [13]. Currently, research is beginning to highlight the importance of microbiome-gut-organ axes in animals. The objective of this review is to highlight the importance of the role the gastrointestinal microbial population plays in modulating the health of a host through communication with different organs in the body through the microbiome-gut-organ axes. Although the microbiome-gut-organ axes are important in terms of all animals and humans, the main focus of this review is how these axes can be utilized in ruminant animals.

## 2. The Gut Microbiota

### 2.1. Development of the Gut Microbiota

Ruminants are born with an undeveloped, nonfunctional rumen; however, the microbial populations begin to establish shortly after birth [3,15]. Despite misconception, ruminants are not born with a sterile gastrointestinal tract. During parturition, a calf is exposed to microbes from the vaginal canal and perineum, which include microorganisms such as *Truperella pyogenes* and species from the genera *Staphylococcus*, *Clostridium*, *Bacteroides*, *Ureaplasma*, and *Mannheimia* [16,17,18,19]. The calf’s microbiota continues to be colonized by microbes derived from the skin during suckling and the oral cavity during licking [17]. This introduces bacteria commonly found on the skin of the udder, *Citrobacter* spp. and *Leuconostoc* spp., which have been detected in the GIT of calves as early as 6 h after birth [20]. In addition to bacterial species being introduced from a calf’s dam, eukaryotic species, protozoa (e.g., *Entodinium*), and fungi (e.g., *Neocallimastix*) found in the GIT are introduced through saliva as the mother licks her calf [21,22].

Another major source of bacterial colonization is colostrum which typically contains species from *Lactobacillus*, *Bifidobacterium*, *Staphylococcus*, *Escherichia coli,* and *Streptococcus uberis* [23]. Of note, some of these bacterial groups are generally classified as opportunistic pathogens that can take advantage of the uninhabited GIT of newborn ruminants. For example, *E. coli* begins to colonize the rumen as early as 24 h after birth [24]. These pathogens can cause gastrointestinal distress early in calves [25]. Previous research detected many opportunistic pathogenic species, including *Escherichia* and *Salmonella,* present in the hindgut of 1-week-old calves, contributing to digestive issues [25].

As a calf’s GIT microbiota matures, there is a change in the composition of the GIT microbiota. The initial pathogenic bacterial species that are generally facultative anaerobes able to utilize oxygen are soon replaced by the more beneficial species that are strict anaerobes [15,26]. With the change in bacteria, the rumen environment becomes anaerobic, with few species, including fungi being able to utilize oxygen. These more beneficial bacterial species (i.e., amylolytic bacteria, lactate utilizers, sulfate-reducing bacteria, and xylan and pectin fermenting bacteria) start out at much lower concentrations but soon dominate the ruminal environment as early as three days of age [3]. The fecal microbiota is dominated by the phyla Bacteroidetes, Firmicutes, and Proteobacteria, with *Bifidobacterium*, *Bacteroides*, and *Lactobacillus* being the most prevalent genera during a calf’s first four weeks of age [27,28,29,30].

### 2.2. Dietary Effects on the Gut Microbiota

As a ruminant’s GIT microbial population continues to establish, diet becomes a key factor contributing to its composition [31,32,33]. When the diet of a ruminant is altered, the GIT microbiota is altered as a result. One of the most dramatic dietary shifts that occurs during the beef production system is the change from a forage-based diet to a concentrate-based feedlot-finishing ration. During this transition, the GIT microbiota becomes unstable resulting in dysbiosis, or an imbalance, allowing GIT distress to occur [34,35]. Many studies have found that the microbial composition of the GIT is altered by age and diet [36,37,38,39]. Previous research revealed that the fecal microbial composition of steers at weaning is different from the microbial composition at yearling and slaughter, which can be attributed to dietary differences [37].

The shift in the microbial composition from a dietary change can affect specific bacterial species with the GIT microbiota. Many bacterial species can be described as either generalists, bacteria able to degrade a wide variety of substrates and thrive in a wide variety of environments, or specialists, bacteria that are only able to degrade a very specific set of substrates that occupy a narrow niche [40]. The abundance of these bacterial species is determined by the nutrients in the diet, which provides a competitive advantage to certain bacteria in the rumen. Earlier in life, when ruminants are fed a predominately forage-based diet, cellulolytic bacteria, including *Ruminococcus albus*, *R. flavefaciens*, and *Fibrobacter succinogenes*, tend to dominate since they are better able to degrade and utilize forages [41,42,43]. The fecal microbial population is made up of mainly the family *Ruminococcaceae* [37], which has many genes present that can bind cellulose, hemicellulose, and xylan, making this family particularly adapted to degrading plant materials [44,45,46]. Once cattle arrive at the feedlot, the diet shifts to a predominately concentrate-based diet and ruminal fermentation must rapidly change as well. Instead of degrading mainly cellulose and hemicellulose, fermentation must shift to mainly degrading starch and soluble sugars [31,47,48]. After this switch to a high concentrate diet, amylolytic bacteria dominate with *Ruminobacter amylophilus*, *Streptococcus bovis*, *Succinomonas amylolytica*, *Butyrivibrio fibrosolvens*, *Selenomonas ruminantium*, and many species from the genus *Prevotella* being to dominate the ruminal niche [49].

### 2.3. Effects on the Immune System

Calves are born with an immature but functional immune system [50]. While in utero, the central organs (e.g., bone marrow and thymus) are fully developed. However, the peripheral organs (e.g., lymph nodes, spleen, and mucosa-associated lymphoid tissues, including the gut-associated mucosal tissue [GALT]), do not fully develop until they are exposed to antigens after birth [51]. The developing GIT microbial population plays a vital role in regulating and activating a calf’s immune system during the early stages of life [17]. Previous research has discovered that the hindgut is crucial for the development of the immune system in monogastric animals [52]. Research is now revealing that the microbial consortium in the hindgut of cattle is equally as important in the development of the calf’s immune system [53], which plays a vital role in the GIT health, feed digestion, and energy production [54].

The epithelial cells in the GIT are joined by an intercellular junction called tight junctions, which are comprised of proteins (e.g., occludin, claudins, zonula occludens) and adhesion molecules. The tight junctions allow the passage of nutrients, ions, and water through the bloodstream while preventing the passage of microbes and their peptides [55,56]. Right after birth, a calf’s tight junctions are not fully formed, and passage into the bloodstream is increased, which improves a calf’s ability to absorb nutrients, immunoglobulins, and leukocytes from the colostrum [57,58]. This permeability begins to decrease around 24 to 36 h after birth and continues to decrease during the calf’s first month [59]. Research has shown that the presence of some bacterial species (e.g., *Lactobacillus* spp. and *Bifodobacterium* spp.) and their metabolites can promote the activation of these tight junction proteins [60,61,62]. This can be very important during the first few weeks of life when the intestinal permeability is just starting to decrease. The colostrum calves receive during the early stages of life plays an important role in host immunity by increasing the hindgut abundance of probiotic species such as Bifidobacterium while decreasing the hindgut abundance of opportunistic pathogenic bacteria *E. coli* and *Escherichia-Shigella* [63]. *Robseburia* and *Oscillospira* have been found to have genes involved in the regulation of host immunity and metabolism, while SCFA receptor genes decrease inflammation and increase intestinal barrier function [14], which is vital during the early stages of development.

Proinflammatory (e.g., interleukin [IL]-8) and anti-inflammatory (e.g., IL-10) cytokines are upregulated during the first week of a calf’s life [64]. These are important because IL-8 works to activate and attract neutrophils into the interstitial space in the body [65], and IL-10 prohibits proinflammatory molecules (e.g., INF-g, TNF-α, IL-6) from activating and prevents the immune system’s ability to recognize antigens [66]. Bacterial species aid the immune system in these functions because *Lactobacillus* and *Bifidobacterium* activate IL-10, which helps prevent the immune system from activating a proinflammatory response to beneficial bacteria within the GIT microbiota [67]. Both *Lactobacillus* and *Bifidobacterium* are introduced to the GIT through colostrum [23], and both play an important role in the early stages of the establishment of the GIT microbiota, which can contribute to controlling the ruminant’s immune response and preventing any unnecessary inflammatory responses to “self” antigens.

Not only does the GIT microbial population aid in establishing the calf’s immune system in the first few weeks of its life, but it also continues to play a vital role in maintaining and regulating the immune system as cattle mature and grow. As calves are transitioned from a milk-based diet to either a forage- or concentrate-based diet, both the calf’s GIT microbiota and the expression of genes related to intestinal immunity are impacted [68]. The expression of immune genes has been linked to the abundance of luminal bacteria and bacteria correlated with GIT in health in dairy calves [69]. IgA, a secretory immunoglobulin that plays a large role in intestinal immunity, aids the immune system in regulating the relationship between both beneficial and pathogenic bacteria [70]. If this immunoglobulin is removed from the GIT, bacteria can increase uncontrollably in abundance while the immune system upregulates the expression of proinflammatory cytokines [71]. Recovery of IgA will restore intestinal homeostasis by returning commensal bacterial species populations and eliminating inflammation [72].

Another key feature of the gastrointestinal system that helps maintain health and proper functioning is mucus, which provides a barrier to tight junctions and aids in maintaining gut integrity [69,73]. Although beneficial for host health, if the production of mucus becomes abnormal, then the GIT can experience distress and disease [73,74]. The GIT microbial population plays a vital role in the production of intestinal mucus. When comparing germ-free and conventionally raised mice, the mice without functional microbiota had less mucus lining their intestinal epithelial cells [75,76]. Meanwhile, when germ-free mice are exposed to bacterial molecules (e.g., lipopolysaccharides and peptidoglycans), their intestinal mucus production is restabilized [75,77]. These results highlight the importance of the GIT microbiota and its metabolites in promoting and maintaining intestinal mucus production.

Due to the communication between the GIT microbiota and the host’s immune system, any disturbance in the equilibrium of the microbiota or the immune system will affect both. Suppose calves are suddenly switched to a different diet and their GIT microbial population is not given time to acclimate. In that case, the expression of claudin and occludin is downregulated in the mucosal barrier of the intestines resulting in an increase in gut permeability [78]. The decreased GIT integrity can be attributed to an increase in the host’s inflammatory response due to an interaction with pathogenic bacteria and their metabolites, or with pro-inflammatory cytokines [68]. Although the immune’s pro-inflammatory response is a defense mechanism designed to target foreign invaders (e.g., pathogens) and works in conjunction with the GIT microbiota to prevent pathogenic invasion and gastrointestinal dysbiosis, the inflammatory response can become too upregulated and negatively impact host health [79]. Another detrimental interaction between the GIT microbiota and the host’s immune system can occur during the onset of rumen acidosis. This digestive issue is caused when the rumen pH drops and becomes acidic as a result of an abundance of rapidly digestible carbohydrates in the diet producing an abundance of lactic acid (acute acidosis) or volatile fatty acids (subacute acidosis) [54]. As a result, the number of cellulolytic bacteria can decrease, and the number of amylolytic bacteria can increase, resulting in the microbiota experiencing dysbiosis [80]. Additionally, as the pH of the rumen decreases, the rumen epithelial cells become damaged, preventing the absorption of nutrients that can ultimately have negative impacts on the host’s performance and health [54]. This highlights the importance of maintaining balance in the microbial populations in the gut and in the host’s immune system to increase growth and health in the host animal.

### 2.4. Therapies to Modulate GIT Health

Since so much interplay exists between the GIT microbial population and the host’s immune system, GIT microbial therapies can be utilized to improve the overall health of a host. Three main types of therapies are utilized to modulate the GIT microbiota—probiotics, prebiotics, and gut microbial transplants. They range from the introduction of a few key species or fermentable products [81,82,83] to a complete functional microbial population [84,85,86,87,88]. Recently, research has focused on how these therapies can be utilized to stabilize the GIT microbiota to prevent diseases [89]. These therapies mainly work by preventing dysbiosis in the GIT microbial community by preventing an increase in harmful bacteria and a decrease in beneficial bacteria, which is especially valuable when the host is undergoing stress.

#### 2.4.1. Probiotics

One of the most commercially available and widely used GIT microbiota therapies available to promote human and animal health is probiotics, or direct-fed microbials (DFM). Probiotics are living microorganisms found naturally in the GIT that have a direct or indirect impact on host health. They can be mono- or mixed cultures usually comprised of bacteria or fungi [83,90]. Probiotics work by producing metabolites that stimulate the growth of commensal bacteria, inhibiting proliferation and colonization of pathogenic bacteria, regulating gastrointestinal pH, promoting mucus production, and improving the function of intestinal epithelial cells [91]. In livestock production, probiotics are valuable tools utilized to improve GIT health, feed efficiency, and milk quality [92,93]. Additionally, they are crucial to preventing dysbiosis from occurring in the GIT microbiota as a result of stressful events such as transportation [94].

In ruminant animals, probiotics can act as alternatives to antimicrobial feed additives by limiting the colonization of pathogenic species. *S. cerevisiae* has been found to improve ruminant production by increasing ruminal pH [95], which is beneficial in both beef and dairy production by preventing acidosis when introducing more starch into the diet. *S. cerevisiae* can aid in nutrient utilization by increasing ruminal fermentation [96]. The introduction of *Lactobacillus* probiotics to the diet of calves has been shown to increase growth and prevent immunocompetence [97]. *Megasphaera elsdenii* and *Butyrivibiro fibrosolvens* have been found to redirect SCFA production from lactate to butyrate, which increases ruminal pH preventing subacute rumen acidosis [98]. Although studies have controversial results on the effects of probiotics in the diet of ruminants, adding probiotics to the diet prior to a stressful event such as weaning or a diet change stabilizes the ruminal ecosystem preventing dysbiosis.

#### 2.4.2. Prebiotics

The practice of introducing substrates that bacteria utilize into the GIT instead of the bacterial species themselves is increasing in popularity. These substrates are called prebiotics because they cause the “prebiotic effect,” which is “the selective stimulation of growth and/or activity of one or a limited number of species in the gut microbiota that confer(s) health benefits to the host” [99]. Prebiotics are comprised of non-starch polysaccharides (NSP) or oligosaccharides. Prebiotics are introduced into the diet to be fermented and utilized by beneficial bacteria to improve the health of the GIT [91]. Therefore, to truly be considered a prebiotic, the product must be indigestible by the host, fermentable by commensal GIT microbiota, and increase the growth of beneficial bacteria in the gut [100,101].

Like probiotics, prebiotics can positively impact cattle production and health. They are usually introduced into ruminant diets alongside probiotics to increase substrates that can be utilized by the microorganisms contained in the probiotic to improve its efficacy [81]. Prebiotics can be fed to increase weight gain and feed efficiency and reduce scours and respiratory diseases [102,103,104]. Fructose oligosaccharides (FOS) have previously been shown to lessen enteric issues in calves [105] and decrease colonization of many pathogenic bacteria, including *Salmonella* and *E. coli* [106]. Adding Galactosyl-lactose (GL) to milk replacers has previously been shown to increase growth while improving overall health status of dairy calves [102]. Prebiotics, also, work best as a preemptive therapeutic to prevent GIT dysbiosis during a stressful time for the host.

#### 2.4.3. Gut Microbial Transplants

The utilization of sequencing technology has provided many recent advances to our knowledge of the role of individual microorganisms within the GIT microbiota. However, all the bacteria comprising the GIT microbiota have yet to be elucidated [107]. Currently, we are only able to culture an estimated 23–40% of the microbes within the rumen [107]; therefore, we are limited in our ability to develop gut therapies like probiotics and prebiotics since we still have not identified every microorganism or their function in the GIT. To overcome our lack of knowledge, gut microbial transplants can be utilized. Within human and small animal medicine, gut microbial transplantation has become a very promising avenue for GIT therapies [84,85,86,87,88]. In terms of cattle production, there can be two types of gut microbial transplants—fecal matter transplants (FMT) and ruminal fluid transplants (RFT).

In ruminant animals, RFT is the most popular form of gut microbial transplants. This introduces rumen fluid from a healthy donor into a recipient experiencing dysbiosis [108]. In sheep experiencing acidosis, an RFT accelerated rumen fermentation, decreased dysbiosis, and repaired damage to the ruminal epithelial cells [109]. An RFT is also a valuable tool prior to weaning in lambs by increasing starch degrading bacteria, thus improving the digestibility of starch-containing diets and increasing propionate production by ruminal microbes [110]. This study also found that an RFT led to faster organ development, especially the hindgut and liver. Although not as commonly used in ruminants, FMT treatment can also be used in cattle. Research has shown an FMT can be an effective treatment for diarrhea [111]. Additionally, this study found that an FMT treatment in a calf’s early life can potentially improve growth performance. Although the idea of inoculation with GIT contents dates back as early as the 1700s [112], research is still needed to fully understand the validity of using an RFT or an FMT as a therapeutic for modulating host health.

## 3. Microbiome-Gut-Organ Axes

Recently, there has been a major drive towards understanding the complex synergistic relationship that exists between the GIT microbial population and the host. The GIT microbiota interacts with all aspects of the body through the different microbiome-gut-organ axes (MGOA). The metabolites produced by the GIT microbiota send signals throughout the body to different organs, which affect the immune system and host physiology [113,114]. This interaction between the microbes colonizing the GIT and the immune system impacts organs throughout the host and forms an “axis” that can send signals [115]. Some examples of these axes in cattle include the established gut-brain axis and gut-lung axis [116] and the proposed gut-mammary axis and gut-reproductive axis (Figure 1).

The different MGOA enable bidirectional communication between the GIT and different organs that occur through signaling pathways [117,118]. The metabolites produced in the GIT by the microbes can have a direct impact on the host’s risk of infection [119]. Dietary or environmental stressors can alter the species diversity within the GIT, leaving the microbiota susceptible to pathogenic colonization [119,120]. Ultimately, any alterations within the gut microbial population can have cascading detrimental effects on the health of the host through these different axes. Therefore, it is imperative we understand the mechanisms of the microbiota within the GIT and its communication with other organs to ensure the health and well-being of animals.

### 3.1. Microbiome-Gut-Brain Axis

One of the most extensively studied gut-organ-axes within all mammalian systems is the microbiome-gut-brain axis (MGBA). This axis serves as a bi-directional link where signals and metabolites can be sent between the brain and the gut. Previous research has revealed the microbes within the GIT play an important role in many processes within the brain of the host, including the development of the brain, neural processes, pain processes, the hypothalamic-pituitary-adrenal (HPA) axis, and behavior [121]. Due to its impact on the brain, the MGBA has been increasing in popularity as a tool to modulate brain function and, ultimately, host health.

There are three different pathways that serve as routes of communication for the MGBA [122]. The first is the immunoregulatory pathway, where the immune system interacts with the microbiota and affects the production of cytokines, cytokinetic reaction factors, and prostaglandin E2 [123], which subsequently alters brain function [13]. The second is the neuroendocrine pathway which involves both the HPA axis and the central nervous system (CNS). The intestines of mammals serve as one of the largest endocrine organs in the body by possessing over 20 different types of enteroendocrine cells [124]. Additionally, the microbes within the gut regulate the production of many neurotransmitters (e.g., cortisol) through the HPA axis and CNS [125]. The last pathway is connected through the vagus nerve and enteric nervous system (ENS). The ENS forms synapses with the vagus nerve, which allows communication between the microbes and the brain [125]. Ruminal fermentation produces metabolites that are toxic to the brain (e.g., ammonia and D-lactic acid), which can travel through this pathway to negatively impact brain function and the host’s stress response and quality of sleep [126]. Additionally, sensory neurons can send signals through the CNS to control gut motility and hormone secretion [13].

The role the MGBA plays in modulating host health has been studied thoroughly in humans; however, research investigating the MGBA has expanded to animals. This communication can occur when there is an infection in the GIT microbiota that negatively impacts the brain and increases sickness behavior or when the GIT microbiota is healthy and promotes brain function. Research utilizing germ-free animals has shown that the inclusion of probiotics and prebiotics results in behavioral changes [127]. Throughout livestock production, there are many stressful events (e.g., weaning and transportation) that are unavoidable. Research has shown that after transportation, there is an increase in cortisol, adrenocorticotropic hormone (ACTH), and pro-inflammatory cytokines (i.e., IL-6, TNF-α, IL-1β) in multiple breeds of beef cattle [128]. This study also found an abundance of ruminal *Lactobacillus* that were positively correlated with IL-6 and IL-4. In addition, probiotics or prebiotics can be added to the diet of ruminants to prevent the signaling of the HPA axis to increase anxiety. A study utilizing dairy calves found supplementing the calf’s diet with a multispecies probiotic prior to weaning improved growth, decreased the incidence of diarrhea, affected the fecal microbiota (e.g., increased abundances of *Bifidobacterium*, *Lactobacillus*, *Collinsella*, and *Saccharomyces*), and reduced serum concentration (i.e., IgA, IgG, and IgM) [129]. Ultimately the impact of the GIT microbiota on behavior in animals directly impacts the overall health of the host since the microbes directly impact the immune system of the host [130].

In addition to general behavior being affected by the GIT microbiota, feed behavior is directly impacted by the microbial population within the GIT, which directly impacts the feed efficiency of livestock. An increase in pathogenic bacteria within the GIT microbiota results in an increase in sickness behavior followed by a reduction in feed intake [127]. When ruminants are fed a high-starch diet, they can experience ruminal acidosis, which can negatively influence feed behavior by decreasing both feed intake and the time spent ruminating [131]. This can be reversed by an RFT by introducing healthy microbes back into the GIT microbiota. This increases feed intake while decreasing inflammation, ultimately promoting host health [132]. In addition to an RFT, the utilization of a probiotic, *S. cerevisiae*, can inhibit the negative effects of ruminal acidosis (e.g., reduced pH) by increasing ruminal pH [92,133]. This influences the MGBA by increasing feed behavior by increasing the time spent ruminating and decreasing the time between feedings [134].

### 3.2. Microbiome-Gut-Lung Axis

Another MGOA discovered more recently is the microbiome-gut-lung-axis (MGLA). Due to the inability to culture microbes from the lungs of healthy hosts, the lungs were originally thought to be sterile unless an infection was present [135]. With the utilization of sequencing technologies (e.g., whole genome sequencing and 16S rRNA gene sequencing), researchers were able to discover that the lungs were inhabited by a community of commensal microbes existing in healthy individuals [136]. These microbial communities play a protective role in the respiratory tract, preventing the colonization of bacteria or viruses, which can cause disease [137]. Much like the GIT microbiota, the respiratory microbiota plays a role in regulating the activation of both the innate and adaptive immune responses [138,139].

The respiratory tract is divided into two parts based on location and function: the upper respiratory tract (URT) and the lower respiratory tract (LRT) [140]. The URT is comprised of the nasal cavity, paranasal sinuses, nasal passages, nasopharynx, oropharynx, tonsils, and upper portion of the larynx. In contrast, the LRT contains the larynx, trachea, bronchi, bronchioles, and alveoli [141]. Much like their function, the URT and LRT are colonized by different microbes [142,143,144] shortly after birth [145]. The most abundant phyla of the LRT are Bacteroidetes and Firmicutes, which is similar to the oral cavity microbiota suggesting the oral cavity plays a role in the development of the lung microbiota [136,146,147]. On the other hand, the microbial population of the nasal cavity is mainly comprised of the phyla Firmicutes and Actinobacteria, which more closely resembles the microbial population of the skin [148,149,150], suggesting the development of the nasal cavity microbiota is influenced by the skin microbiota. Due to differences in the suspected sources of colonization of the URT and LRT, researchers need to be cautious when drawing conclusions about the microbiota of one location based on the other. However, research has revealed correlations between microbes within the URT and LRT that can influence both microbial communities [144]. Other factors are known to influence the development of the respiratory microbiota, including diet [151], genetics, age [145], vaccination administration, management, and environment [152,153].

Although signaling through the MGLA travels bi-directionally, the majority of the communication between the microbial populations of the GIT and respiratory tract travels from the gut to the lungs [135]. The specific mechanisms and pathways involved in the MGLA in cattle remain undiscovered. Still, micro-aspiration, inhalation of bacteria, and transfusion of bacteria through mucosal cells play an important role in the communication [154]. Additionally, the lymphatic system and bloodstream play an important role in communication between the GIT and respiratory tract by carrying bacteria and bacterial metabolites from the GIT to the lungs [155].

Within the respiratory tract, the nasopharyngeal mucosal layer serves as the first line of defense against pathogenic colonization by capturing particles that are inhaled through the respiratory tract and moving them back up into nasal and oral cavities [141]. The mucus contains immune cells, including antimicrobial peptides, glycoproteins, and IgA, that help maintain homeostasis in the respiratory microbiota [156,157]. The second line of defense is the mucosal epithelium which produces molecules that trigger innate and adaptive immune responses to improve barrier function [158,159,160,161]. Not only do the respiratory tract epithelial cells, but luminal and mucosal surface macrophages and dendritic cells express innate patter-recognition receptors that work to identify and clear pathogenic microorganisms [161,162]. The commensal microbes comprising the microbiota of the respiratory tract, mucosal epithelium, and the immune system communicate to promote respiratory health, reduce inflammation, and maintain a functioning microbiota [141].

One of the main ways the GIT microbiota influences the immune system and, thus, the respiratory tract through the MGLA is with SCFA [135]. SCFA are important for maintaining intestinal integrity and preventing inflammation in both the gut and the respiratory tract [163,164,165]. SCFA can enhance the intestinal epithelial barrier function within the gut by increasing mucus production by goblet cells [166,167] and strengthening tight junctions [168]. Additionally, SCFA increase IgA production by enhancing plasma B cells metabolism to ensure the intestines are protected from inflammation [169,170]. An individual SCFA, butyrate, aids the intestinal epithelium in suppressing inflammation to maintain homeostasis [171]. SCFA also promote intestinal homeostasis through a positive feedback loop directing the metabolism of intestinal cells toward increased fatty acid ß-oxidation [172]. Butyrate supplementation has the ability to improve epithelial integrity while also increasing the host’s defense mechanisms [173].

Due to the numerous stressors calves are exposed to during weaning, it serves as one of the most influential times for respiratory microbiota development [151,152,174,175]. The composition of the URT is majorly affected by the first 40 days after arrival at the feedlot [176]. This is due to the stress, exposure to diseases, and dietary changes that occur when calves arrive at the feedlot that can result in dysbiosis in the URT, which weakens a calf’s immune response and allow pathogens in the URT to migrate into the LRT [146,154,177,178]. In dairy calves experiencing illness, there was an increase in *Mannheimia, Moraella,* and *Mycoplasma* in their URT compared to that of healthy calves [179]. Additionally, the URT of calves that later developed pneumonia was inhabited by a greater number of bacteria at three days of age than calves that remained healthy.

Bovine raspatory disease (BRD) is one of the most significant health concerns that can occur in weaned calves or feedlot cattle shortly after transportation [180]. After years of research to eradicate the disease, BRD remains one of the leading causes of morbidity, mortality, welfare issues, and economic losses within beef production [181]. It is influenced by a combination of factors, including the host, environment, and management [175,180,182]. The bacteria causing BRD are common commensals (e.g., *Mycoplasma*, *Mannheimia*, *Histophilus*, and *Pasteurella*) within the URT microbiota of healthy and sick cattle that can translocate from the URT to the LRT through inhalation after the host undergoes stress resulting in the development of pneumonia [177,179,183,184,185]. One example of how these commensals cause disease is in the case of *Mannheimia haemolytica*. After the stressors occurring after arrival at a feedlot occur and cause dysbiosis, *M. haemolytica* rapidly proliferates within the URT [186]. It then travels to the bronchial epithelial cells, where it damages tight junction proteins, causes lesions in the lungs, releases leukotoxins and lipopolysaccharides, which cause further damage to the respiratory tract, and triggers the host’s immune response causing inflammation. Due to this disease occurring as a result of a stressful event to the host, the MGLA may play a role in the development of this disease; therefore, the GIT microbiota may aid producers in helping eradicate the disease.

Several studies have shown that employing management strategies can positively impact the nasopharyngeal microbial diversity in calves post-weaning [141]. Management strategies can thus help the immune system maintain respiratory health during stressful times throughout the production cycle that leaves the host susceptible to diseases. Research has shown that dietary changes that influence the GIT microbiota shortly after weaning can also have an impact on the respiratory microbiota [151]. One example is preconditioning weaned calves for nine weeks prior to entry into a feedlot with selenium-fortified alfalfa hay can positively impact the microbial population in the nasal cavity [151]. This suggests fortifying the GIT microbiota prior to stress can help prevent respiratory disease through the MGLA.

### 3.3. Microbiome-Gut-Mammary Axis

As research on the GIT microbiota continues to progress, it becomes evident that the microbial population plays a major role in the functioning and disease prevention in other organs besides the brain and the lungs. Within cattle production, the mammary gland is an organ with major importance not only for cattle in general but also for producers in terms of milk production. Much like the GIT and the respiratory tract, the mammary system is also inhabited by a community of microbes [187]. Research into the milk microbiota has begun to suggest communication between the GIT microbiota and the mammary microbiota through a microbiome-gut-mammary axis (MGMA).

The microbiota of the milk is influenced by direct and indirect contact. Direct contact comes from contact with the surface of the teat from milking machines or other dairy equipment [188]. Indirect contact stems from environmental factors, including bedding, feces, forage, drinking water, washing water, air, and the milker itself [189,190]. A study of healthy Holstein Friesian and Rendena cows showed the milk microbiota was influenced by breed [188]. Although there were differences present, Firmicutes was the most abundant phylum, with *Streptococcus* being the most abundant genus. Milk from cows infected with mastitis was dominated by the order Enterobacteriales, followed by Pseudomonadales, Bacillales, and Lactobacillales [191]. In the purebred cattle, *E. coli* was the most abundant genus, followed by *Pseudomonas aeruginosa*, *P. mendocina*, *Shigella flexneri*, and *Bacillus cereus*; whereas, in the crossbred cattle, the most dominant genus was *Staphylococcus aureus* followed by *Klebsiella pneumoniae*, *S. epidermidis*, and *E. coli*. Although the majority of the bacteria detected are pathogenic, there are still other commensal strains present. Previous studies have found some of the commensals can come from direct contact with skin or can travel from the intestinal lumen to the mammary gland through the communication of the MGMA [187,192].

The mechanisms by which the mammary gland and the GIT microbiota communicate with each other has yet to be fully elucidated. The current knowledge of the route of communication between these two organs is from research done in humans and mice. A previous study reported that bacteria travel from the GIT to the mammary system via dendritic cells and macrophages [193]. These cell types, which are in the GALT, can internalize bacteria from the GIT microbiota and translocate it to a different location, such as the mammary gland [194,195]. This suggests an infection within the mammary system may affect the integrity of the epithelial cells allowing translocation of pathogenic bacteria through the MGMA.

Within dairy production, mastitis is the most impactful disease for dairy producers [196]. It causes a major economic loss worldwide by decreasing both the quality and quantity of milk [197]. A study of healthy and infected dairy cows revealed the microbiota of milk from infected quarters has more variation compared to the milk microbiota of healthy quarters, suggesting the milk microbiota experiences dysbiosis when infection occurs [198]. When a quarter is infected, the microbial community can become dominant by a specific microbial taxon. When comparing milk and skin microbiotas from cows with mastitis, *Staphylococcus* spp. and *Debaryomyces* spp. were shared between them [198], indicating a link between the skin and milk microbiota.

There is debate on the validity of mammary microbiota, with some theories speculating bacteria sequenced from the mammary gland being contamination introduced during sample collection, sample processing, or DNA extraction [199,200]. When determining if bacteria isolated from the mammary gland is part of a community of microorganisms or a pathogen infecting a sterile environment, researchers need to objectively evaluate the validity of their results to determine if the differences are biological in nature, if the relationship is causative, what the mechanism of action is, how effective the experimental design was at answering the hypothesis, and if any potential factors could be contributing to the experimental differences [201]. Research has shown sequencing samples using different methods can result in differences [202], so it is imperative to maintain proper sterile techniques to ensure accuracy when performing microbiome studies [107]. However, despite this debate and the many theories surrounding it, the mammary gland can still be impacted by the GIT microbiota through the MGMA. This impact may be direct by influencing the colonization of the mammary gland or indirect by the GIT causing systemic inflammation after gut dysbiosis [68], which can lead to inflammation in the mammary system and potential for disease.

### 3.4. Microbiome-Gut-Reproductive Axis

Like the MGMA, the relationship that exists between the GIT microbiota and the reproductive system remains relatively unexplored. Much like the GIT microbial population, the reproductive microbiota can experience dysbiosis where a disease state occurs. Research has begun to explore the interactions that exist between the GIT microbiota and the reproductive microbiota, which can be referred to as the microbiome-gut-reproductive axis (MGRA), and its role in preventing pathogenic colonization and promoting host health. With the importance of reproductive health for both beef and dairy production, the MGRA could majorly impact the reproductive efficiency of cattle.

For a long period of time, much like the lungs, the reproductive tract of healthy females was considered sterile [203,204,205]. With recent advances in sequencing technologies, there has been increasing evidence that the reproductive tract of humans and animals is inhabited by a resident microbiota [206]. This microbial population begins to colonize the reproductive tract shortly after birth. At birth, the barrier of the cervix is compromised, which can allow microbes to travel from the vagina, the environment, feces, or skin into the reproductive tract [207,208].

The different sections of the reproductive tract have distinct microbial populations; however, they can still influence each other. The vaginal niche is mainly comprised of the phyla Firmicutes, Bacteroidetes, and Proteobacteria [18,209]. A study of *Bos Indicus* breeds found the most abundant bacterial genera in the vagina were *Aeribacillus*, *Bacillus*, *Clostridium*, *Bacteroides*, and *Ruminococcus* [210]. Very similarly, the most dominant phyla in the cervix have been found to be Proteobacteria, Bacteroidetes, and Firmicutes [211]. Meanwhile, the uterus has been found to be colonized by the phyla Proteobacteria, Tenericutes, Firmicutes, Bacteroidetes, Fusobacteria, and Actinobacteria [212].

Previous research has highlighted the bloodstream as a major route of communication for the MGRA [213]. This study found the bloodstream contained a microbiota that was colonized with the main pathogens within the uterine microbiota, such as *Bacteroides*, *Porphyromonas*, and *Fusobacterium,* which are found in the blood shortly after calving, suggesting the bloodstream has a role in bacteria translocation. Additionally, some bacteria which cause liver abscesses in cattle, *T. pyogenes* and *F. necrophorum*, have been isolated in the uterus, suggesting these bacteria travel from the liver to the uterus through the bloodstream [214]. The hematogenous pathway can be utilized to carry bacteria from the GIT microbiota to the reproductive tract microbiota; thus, the GIT microbial population influences the composition of the vaginal and uterine microbial populations [18]. For example, *Bacteroides heparinolyticus*, identified in both the feces and blood of dairy cows, is assumed to be part of the uterine microbiota.

Through the MGRA, the GIT microbiota plays a major role in the functioning of the reproductive system. Research in dairy cows found a genetic similarity in strains of *E. coli* found in the GIT and uterus, suggesting the GIT aids in the colonization of the uterine microbiota through ascending colonization of bacteria by the lower genital tract [215]. Butyrate supplementation in postpartum cows positively influenced breeding capacity by re-establishing estrous and restoring ovarian function earlier than in supplemented cows [216]. The ratio of circulating follicle-stimulating hormone (FSH)/luteinizing hormone (LH) has been positively correlated with systemic lipopolysaccharide (LPS) and the bacterial genera *Actinobacteria*, *Bacteroides*, and *Streptococcus* [217]. A study in male sheep found that supplementing the diet with SCFA, including acetate, propionate, and butyrate, increased the secretion of LH and FSH [218]. Previous research in dairy cows found the uterine and vaginal microbiotas were most similar in composition seven days postpartum [219]. Additionally, cows that later developed endometritis at 21 days postpartum had a delay in uterine and vaginal microbiota differentiation and a decrease in bacterial diversity at seven days postpartum. The presence of *Bacteroides, Poryphomonas,* and *Fusobacteria* was associated with metritis occurring after a decrease in bacterial richness, causing uterine dysbiosis [220]. This indicates these two environments may indicate future reproductive diseases. Specific bacterial families (*Lachnospiraceae* and *Rikenellaceae*) and genera (*Acinetobacter*, *Bacillus*, *Oscillospira*, CF231, and 5–7NS) have been identified as an indication of a healthy vaginal microbiota and have the potential to be a therapeutic target [221]. Research has demonstrated that the MGRA may be a valuable tool to improve reproductive efficiency in cattle herds by preventing reproductive diseases and increasing hormone secretion.

## 4. Conclusions

The many microbiome-gut-organ axes throughout a host highlight the importance of the GIT microbial populations in regulating homeostasis by preventing colonization of pathogenic species, thus decreasing disease/infection prevalence. By stabilizing the GIT microbiota, all organ systems in the host can function at optimal levels. These axes can therefore be a novel target for therapeutics to prevent certain diseases or infections. Although this research has come a long way in the past couple of years, there is still much that remains unknown about the microbiome-gut-organ axes, specifically in ruminants. As improvements are made in sequencing techniques, we can gain a more comprehensive understanding of all the microbes with the GIT and other organs. Additionally, with advances, we can further elucidate the roles the microbes play in regulating the host immune system as well as the numerous organ systems throughout the body. With the addition of future research, producers will be able to target the many MGOA to increase overall health and efficiency in their herds.

## Figures and Tables

**Figure 1 microorganisms-10-01391-f001:**
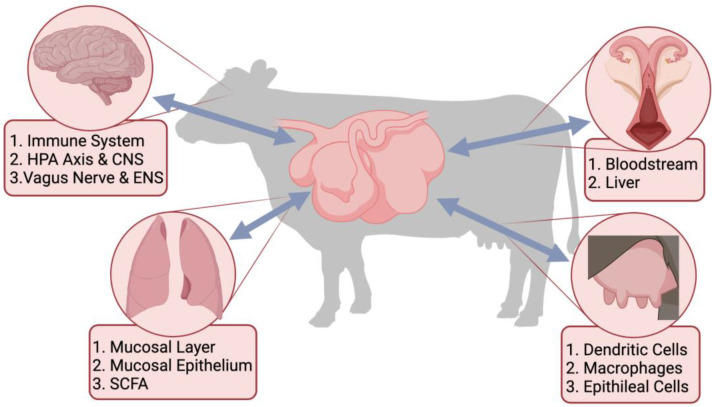
Proposed links between the gastrointestinal tract microbiota and different organ systems through the microbiome-gut-organ axes, including the microbiome-gut-brain axis (MGOA), the microbiome-gut-lung axis, the microbiome-gut-reproductive axis, and the microbiome-gut-mammary axis. Included are pathways, cells, and metabolites important for the bi-directional communication of the MGOA.

## Data Availability

Not applicable.

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
