# Peer review of "Utilizing the Gastrointestinal Microbiota to Modulate Cattle Health through the Microbiome-Gut-Organ Axes"

_microorganisms, 2022, doi:10.3390/microorganisms10071391_

Round 1
Reviewer 1 Report
My comments on the manuscript are as follows:
1. Title: In the “main body” of the article, the authors refer exclusively to cattle. So why in the title it is mentioned generally “Ruminant”. The title should be adequate to the content of the article. Please consider to revise it.
2. L. 69 – 72: “eukaryotes, protozoa, and fungi….”. Please mention the main representatives.
3. L 99-100: “This shift in the microbiota can affect specific bacterial species with the GIT microbiota”. Which shift and which specific bacteria species effect?
4. L 168-175: Okay for germ free mice, but is there any literature or studies on ruminants?
5. L 232-235: “There is increasing use of introducing bacterial metabolites into the GIT instead of the bacterial species themselves. This causes what is known as the “prebiotic effect” which is “the selective stimulation of growth and/or activity(ies) of one or a limited number of species in the gut microbiota that confer(s) health benefits to the host””. Are the bacterial metabolites prebiotics? Please revise it.
6. L 320-321: “The intestines of mammals serve as one of the largest endocrine organs in the body by producing over 20 different types of cells”. Please give an explanation of what is mentioned.
7. The majority of the literature in section 3.2 is about humans and I do not think it is wise to take ownership of the case of cattle or ruminants.
8. L 553-572: I do not think this paragraph belongs to this section.
9. Section 3.3 Microbiome-Gut-Mammary Axis. The authors use studies on microbiological issues of milk hygiene to document that “the mammary system is also inhabited by a community of microbes”, and I do not think that is right.
Mammary microbiota of dairy ruminants: fact or fiction? Rainard P. Mammary microbiota of dairy ruminants: fact or fiction?. Vet Res. 2017;48(1):25. Published 2017 Apr 17. doi:10.1186/s13567-017-0429-2.
- Please discuss the methodological limitations of the article.
Author Response
- Title: In the “main body” of the article, the authors refer exclusively to cattle. So why in the title it is mentioned generally “Ruminant”. The title should be adequate to the content of the article. Please consider to revise it.
The title has been updated to now read just cattle to reflect the reviewer’s suggestion.
- L. 69 – 72: “eukaryotes, protozoa, and fungi….”. Please mention the main representatives.
Based on the reviewer’s comment, the following additions have been added, “…eukaryotic species, protozoa (e.g., Entodinium) and fungi (e.g., Neocallimastix), found…”
- L 99-100: “This shift in the microbiota can affect specific bacterial species with the GIT microbiota”. Which shift and which specific bacteria species effect?
In order to clarifying the sentence, it has been edited to now say, “The shift in the microbial composition occurring from a dietary change can effect specific bacterial species with the GIT microbiota.”
- L 168-175: Okay for germ free mice, but is there any literature or studies on ruminants?
Although it would be more valuable to see the difference a GIT microbiota makes in ruminants, we unfortunately do not currently have a way to develop ruminants that are sterile. Due to this, the only way we can know the true impact of a microbiota is to cautiously draw conclusions from other species.
- L 232-235: “There is increasing use of introducing bacterial metabolites into the GIT instead of the bacterial species themselves. This causes what is known as the “prebiotic effect” which is “the selective stimulation of growth and/or activity(ies) of one or a limited number of species in the gut microbiota that confer(s) health benefits to the host””. Are the bacterial metabolites prebiotics? Please revise it.
Based on the reviewer’s confusion, the authors have edited these sentences to further explain what they mean. The section added says,” There is increasing use of introducing substrates that can be utilized by bacteria into the GIT instead of the bacterial species themselves. These substrates are called prebiotics because they cause what is known as the “prebiotic effect” which…”
- L 320-321: “The intestines of mammals serve as one of the largest endocrine organs in the body by producing over 20 different types of cells”. Please give an explanation of what is mentioned.
Since this sentence was causing some confusion, the following has been added to clarify what the cells are, “The intestines of mammals serve as one of the largest endocrine organs in the body by possessing over 20 different types of enteroendrine cells.”
- The majority of the literature in section 3.2 is about humans and I do not think it is wise to take ownership of the case of cattle or ruminants.
The authors agree with the reviewer and acknowledge the limitations that arise from including examples only from humans or nonruminants. Due to this we included more information derived from cattle research in this section as well as throughout the review to further strengthen the story and conclusions made.
- L 553-572: I do not think this paragraph belongs to this section.
Based on the reviewer’s comment, this paragraph was removed. The sentences about ruminants was added to the paragraph before and additional sentences were added. The paragraph now reads, “Through the MGRA, the GIT microbiota plays a major role in the functioning of the reproductive system. Research in dairy cows found a genetic similarity in strains of E. coli found in the GIT and uterus suggesting the GIT aids in colonization of the uterine microbiota through ascending colonization of bacteria by the lower genital tract (Jones et al., 2022). Butyrate supplementation in postpartum cows positively influenced breeding capacity by re-establishing estrous and restoring ovarian function earlier than in supplemented cows [208]. The ratio of circulating FSH/ LH has been positively correlated with systemic lipopolysaccharide (LPS) and the bacterial genera Actinobacteria, Bacteroides, and Streptococcus [214]. A study in male sheep found that supplementing the diet with SCFA including acetate, propionate, and butyrate increased the secretion of LH and FSH [215]. Previous research in dairy cows found the uterine and vaginal microbiotas were most similar in composition 7 days postpartum (Miranda-CasoLuengo et al., 2019). Addition-ally, cows that later developed endometritis at 21 days postpartum had a delay in uterine and vaginal microbiota differentiation and a decrease in bacterial diversity at 7 days postpartum. The presence of Bacteriodes, Poryphomonas, and Fusobacteria were associated with metritis occurring after a decrease in bacterial richness causing uterine dysbiosis (Galvao et al., 2019). This indicates these two environments may indicate future reproductive diseases. Specific bacterial families (Lachnospiraceae and Rikenellaceae) and genera (Acinetobacter, Bacillus, Oscillospira, CF231, and 5-7NS) have been identified as an indication of a healthy vaginal microbiota and have the potential to be a therapeutic target (Moreno et al., 2021). Research has demonstrated the MGRA may be a valuable tool to improve reproductive efficiency in cattle herds by preventing reproductive diseases and increasing hormone secretion.”
- Section 3.3 Microbiome-Gut-Mammary Axis. The authors use studies on microbiological issues of milk hygiene to document that “the mammary system is also inhabited by a community of microbes”, and I do not think that is right.
Mammary microbiota of dairy ruminants: fact or fiction? Rainard P. Mammary microbiota of dairy ruminants: fact or fiction?. Vet Res. 2017;48(1):25. Published 2017 Apr 17. doi:10.1186/s13567-017-0429-2.
The reviewer’s concerns have been addressed with the inclusion of… “There is debate on the validity of a mammary microbiota with some theories speculating bacteria sequenced from the mammary gland being contamination introduced during sample collection, sample processing, or DNA extraction (Hillerton, 2020; paper they cited). When determining if bacteria isolated from the mammary gland is part of a community of microorganisms or a pathogen infecting a sterile environment, researchers need to objectively evaluate the validity of their results to determine if the differences are biological in nature, if the relationship is causative, what the mechanism of action is, how effective the experimental design was at answering the hypothesis, and if any potential factors could be contributing to the experimental differences (Hanage, 2014). Research has shown sequencing samples using different methods can result in differences (Taponen et al., 2019), so it is imperative to maintain sterile proper techniques to ensure accuracy when performing microbiome studies (Lourenco and Welch, 2022). However, despite this debate and the many theories surrounding it, the mammary gland can still be impacted by the GIT microbiota through the MGMA. This impact may be direct by influencing the colonization of the mammary gland or indirect by the GIT causing systemic inflammation after gut dysbiosis [66] which can lead to inflammation in the mammary system and a potential for disease.”
Please discuss the methodological limitations of the article.
The authors acknowledge there are current limitations in our understanding of the role of the GIT microbiota plays in maintaining the functioning of the host’s immune system. In the conclusion, the following has been added to highlight these limitations, “As improvements are made in sequencing techniques, we can gain a more comprehensive understanding of all of the microbes with the GIT and other organs. Additionally, with advances, we can further elucidate the roles the microbes play in regulating the host immune system as well as the numerous organ systems throughout the body.”
Reviewer 2 Report
I revised the manuscript titled "Utilizing the gastrointestinal microbiota to modulate ruminant health through the microbiome-gut-organ axes" the overall idea of the manuscript is relevant for the field. However, there are many imprecisions across the manuscript. The authors did not provide a concise explanation from the microbiome standpoint. Most of the mechanisms provided need to be rephrased and explained from the microbial standpoint. The authors should not use mice and non-ruminant research to explain some of these mechanisms because there are significant differences between ruminants and non-ruminants. Finally, the manuscript has many typos (parenthesis, incorrect formatting, and missing period) that the authors should carefully revise before submission. Thus, a substantial revision is required before further consideration for publication. Please see the comments below and the references provided.
L26 Please rephrase the sentence “which acts as a fermentation chamber to degrade feedstuffs and absorb organic acids before they reach the glandular stomach”
L29 Produce products are redundant, please correct the whole sentence
L31 is the site in which more than 85%...
L34 remove the word crude
L39 Please include the concept of cross-feeding in this explanation
L43 The relationship between the host and the microbes is not only mutualistic but also symbiotic, please rephrase
L46-L48 Please provide references to support this statement, including studies that evaluated changes in the rumen microbiome associated with the immune response
L55 Please remove the human microbiome, this review focused on the rumen microbiome
L58 Production efficiency has no relationship with the topic discussed in this review, please remove it
L78 The citation style is not correct, please correct Song et al. (). Please revise thorough the manuscript.
L81 Please correct the paragraph describing the transition from facultative anaerobes to strictly anaerobes in young ruminants. There is a well-known transition between these two groups of microbes including protozoa and fungi.
L89 Diet is not an environmental factor, please correct
L96-L98 This statement is unclear, please correct the statement and see my comments in L78
L99 Please revise the sentence
L100-101 This statement is not correct, there are more than two classes of rumen microbes Anaerovibrio lipolytica is just one example. To further explain these differences, the authors must explain the differences between generalist bacteria and specialist bacteria (see https://doi.org/10.3389/fmicb.2015.00296)
L139 Please expand the role of colostrum on the hindgut microbiome and the subsequent development of the immune system in newborn animals. (See https://doi.org/10.1093/femsec/fiy203 and https://doi.org/10.1038/s41396-021-00925-x)
L176- L178. This paragraph is confusing, please rephrase it
L189 Change gastrointestinal pH for rumen pH, the complete name is rumen acidosis. Please correct the paragraph
L191 This statement is not accurate, please revise in detail the effects of rumen acidosis in microbial populations in cattle (See https://doi.org/10.3390/ani9050232)
L200-L202 This statement is not clear, please rephrased it
L225 The use of s. cerevisiae is not only limited to preventing acidosis but also improving rumen fermentation and immune status. Please correct and improve the statement by adding the corresponding references (See https://doi.org/10.1186/s40104-019-0419-5)
L229 Please include the roles of megasphaera elsedii as a DFM to prevent rumen acidosis and to promote propionate synthesis in the rumen. (10.3389/fmicb.2019.00162)
L252-L 256 This statement is very informal and needs to be corrected, please avoid vague statements like “there is still much to learn”, and please provide information supported by references.
L263 Please revise the sentence
L267 The citation style is not correct, please correct
L272-L280 This paragraph has little relevance for ruminants, please add and include information related to ruminants or remove the paragraph.
L305 Remove humans
L335-L340 Please remove the examples with chicks and horses and provide examples of how transport stress and weaning stress influenced both the microbiome and immune status in ruminants
(https://doi.org/10.3390/ani9090599, 10.3389/fmicb.2021.681014)
L341 Avoid informal language (so)
L352 negatively influence
L356 the negative effects of rumen acidosis by (please describe the mode of action).
L358-L360 This statement is not valid, please remove it
L362-L364 in humans or animals? Please clarify
L376 what microbes, please use concise language
L395 This sentence needs to explain the connection between the microbiome and the respiratory tract and the GIT tract.
L406 What microbes? Please provide examples
L419 What about ruminants? Organic acids like butyrate promote the growth and development of rumen papillae and prevent inflammation in the rumen. Please delete this sentence a provide examples with ruminants.
L422-L423 This paragraph is unclear, please correct it
L426-L429 Please mention some of the bacterial or microbial communities observed across the respiratory tract and remove this paragraph. (See: 10.1038/srep29050)
L430-L439 This paragraph should explain the causes of BRD from the microbial standpoint providing examples of commensal microorganisms in the respiratory tract and their changes when the disease occurs. ( See https://doi.org/10.1186/s13567-021-01020-x)
L441 This information is important from the physiological standpoint but with little relevance from the microbiome perspective. The information presented in this review should focus on the microbial aspects and physiological (https://doi.org/10.1186/s40168-020-00869-y)
L499 There are better examples of mastitis with dairy ruminants, please correct this paragraph and provide a more detailed explanation. See 10.1038/s41598-017-08790-5 and https://doi.org/10.1371/journal.pone.0225001
L515 Please revise the sentence and delete the parenthesis (check the whole manuscript)
L543-L546 Please avoid using examples with mice and non-ruminants, there are examples with ruminants (Please see https://doi.org/10.1016/j.vetmic.2022.109355)
L553-L572 Please explain the differences in the vagina microbiome between healthy ruminants compared to sick ruminants. (See https://doi.org/10.1016/j.rvsc.2021.11.007, https://doi.org/10.3168/jds.2019-17106, https://doi.org/10.1371/journal.pone.0200974 )
Author Response
L26 Please rephrase the sentence “which acts as a fermentation chamber to degrade feedstuffs and absorb organic acids before they reach the glandular stomach”
This sentence has been changed to reflect the reviewer’s comment.
L29 Produce products are redundant, please correct the whole sentence
To remove the redundancy, “products” has been changed to “metabolites.”
L31 is the site in which more than 85%...
Changed to reviewer’s suggestion.
L34 remove the word crude
Removed due to reviewer’s suggestion.
L39 Please include the concept of cross-feeding in this explanation
Based on the reviewer’s comment to clarifying this process, “… in a process known as cross-feeding” was added to the end of this sentence.
L43 The relationship between the host and the microbes is not only mutualistic but also symbiotic, please rephrase
Due to the reviewer’s comment, the original sentence has been rewritten to read, “The symbiotic relationship existing between microorganisms colonizing the GIT and the host is mainly mutualistic in nature.”
L46-L48 Please provide references to support this statement, including studies that evaluated changes in the rumen microbiome associated with the immune response
A citation has been added at the reviewer’s request.
L55 Please remove the human microbiome, this review focused on the rumen microbiome
The part about humans was removed and now the sentence reads “Currently, research is beginning to highlight the importance of microbiome-gut-organ axes in animals.”
L58 Production efficiency has no relationship with the topic discussed in this review, please remove it
The part about production efficiency has been removed based on the reviewer’s comment.
L78 The citation style is not correct, please correct Song et al. (). Please revise thorough the manuscript.
The citation has been corrected to match the journals style.
L81 Please correct the paragraph describing the transition from facultative anaerobes to strictly anaerobes in young ruminants. There is a well-known transition between these two groups of microbes including protozoa and fungi.
This section has been expanded to introduce the idea of oxygen being present in the rumen in the beginning and it eventually becoming anaerobic. The following statements have been added to hopefully increase the knowledge presented in this introductory section, “As a calf’s GIT microbiota matures, there is a change in the composition of the GIT microbiota. The initial pathogenic bacterial species that are generally facultative an-aerobes able to utilize oxygen are soon replaced by the more beneficial species that are strict anaerobes [14, 25]. With the change in bacteria the rumen environment becomes anaerobic with few species including fungi being able to utilize oxygen.”
L89 Diet is not an environmental factor, please correct
The word environmental was removed.
L96-L98 This statement is unclear, please correct the statement and see my comments in L78
Based on the reviewer’s suggestion, the sentence has been altered to help clarify its meaning and now says, “Previous research revealed the fecal microbial composition of steers at weaning is different from the microbial composition at yearling and slaughter, which can be attributed to dietary differences [36].”
L99 Please revise the sentence
The following sentence has been revised and now says, “The shift in the microbial composition occurring from a dietary change can effect specific bacterial species with the GIT microbiota.”
L100-101 This statement is not correct, there are more than two classes of rumen microbes Anaerovibrio lipolytica is just one example. To further explain these differences, the authors must explain the differences between generalist bacteria and specialist bacteria (see https://doi.org/10.3389/fmicb.2015.00296)
Based on the reviewer’s comment, the sentence has been changed to correct it and an additional sentence was added. It now states, “Many bacterial species can be described as either generalist, bacteria able to degrade a wide variety of substrates, or specialists, bacteria able to degrade a very specific set of substrates very well (Weimer, 2015). Additionally, bacteria can be classified as either cellulolytic or amylolytic based on the main feedstuff they degrade [3].”
L139 Please expand the role of colostrum on the hindgut microbiome and the subsequent development of the immune system in newborn animals. (See https://doi.org/10.1093/femsec/fiy203 and https://doi.org/10.1038/s41396-021-00925-x)
Based on the reviewer’s comments, the following has been added to further expand this section, “The colostrum calves receive during the early stages of life plays an important role in host immunity by increasing the hindgut abundance of probiotic species such as Bifidobacterium while decreasing the hindgut abundance of opportunistic pathogenic bacteria E. coli and Escherichia-Shigella (Song et al., 2018). Robseburia and Oscillospira have been found to have genes involved in regulation host immunity and metabolism while SCFA receptor genes decrease inflammation and increase intestinal barrier function (Fan et al., 2021) which is vital during early stages of development.”
L176- L178. This paragraph is confusing, please rephrase it
It has been revised to clarify it for the readers and now says. “Due to the communication between the GIT microbiota and the host’s immune system, any disturbance in the equilibrium of the microbiota or the immune system will result in both being affected.”
L189 Change gastrointestinal pH for rumen pH, the complete name is rumen acidosis. Please correct the paragraph
The correction of gastrointestinal to rumen and ruminal to rumen have been implemented into the paragraph.
L191 This statement is not accurate, please revise in detail the effects of rumen acidosis in microbial populations in cattle (See https://doi.org/10.3390/ani9050232)
In order to improve the current sentence it has been revised and now reads, “As a result, cellulolytic bacrteria can experience a reduction which can result in the microbiota experiencing dysbiosis (Ogunade et al., 2019).”
L200-L202 This statement is not clear, please rephrased it
In order to clarify this sentence, it has been revised and now reads, “They range from the introduction of a few key species or fermentable products [78-80] to a complete functional microbial population [81-85].”
L225 The use of s. cerevisiae is not only limited to preventing acidosis but also improving rumen fermentation and immune status. Please correct and improve the statement by adding the corresponding references (See https://doi.org/10.1186/s40104-019-0419-5)
Based on the reviewer’s suggestion, the following sentence has been added, “S. cerevisiae can aid in nutrient utilization by increasing ruminal fermentation (Adeyemi et al., 2020).”
L229 Please include the roles of megasphaera elsedii as a DFM to prevent rumen acidosis and to promote propionate synthesis in the rumen. (10.3389/fmicb.2019.00162)
Based on the reviewer’s suggestion, the following sentence has been added, “Megasphaera elsdenii and Butyrivibiro fibrosolvens have been found to redirect SCFA production from lactate to butyrate which increases ruminal pH preventing subacute rumen acidosis (Chen et al., 2019).”
L252-L 256 This statement is very informal and needs to be corrected, please avoid vague statements like “there is still much to learn”, and please provide information supported by references.
Based on the reviewer’s comment, the sentence has been edited to include a reference and clearer language is utilized. It now reads, “The utilization of sequencing technology has provided many recent advances to our knowledge of the role of individual microorganisms within the GIT microbiota; however, all of the bacteria comprising the GIT microbiota have yet to be elucidated (Lourenco and Welch, 2022).”
L263 Please revise the sentence
Based on the reviewer’s comment, this sentence has been revised and now states, “In sheep experiencing acidosis, a RFT accelerated rumen fermentation, decreased dysbiosis, and repaired damage to the ruminal epithelial cells.”
L267 The citation style is not correct, please correct
Citation style was fixed to match the journal’s style.
L272-L280 This paragraph has little relevance for ruminants, please add and include information related to ruminants or remove the paragraph.
In order to remove parts about humans and nonruminants, the paragraph was removed, and the following was added to the previous paragraph, “Although not as commonly used in ruminants, FMT treatment can also be used in cattle. Research has shown a FMT can be an effective treatment for diarrhea (Kim et al., 2021). Additionally, this study found that a FMT treatment in a calf’s early life can potentially improve growth performance. Although the idea of inoculation with GIT contents dates back as early as the 1700s [105], research is still needed to fully understand the validity of using a RFT or a FMT as a therapeutic for modulating host health.”
L305 Remove humans
Based on the suggestion, the part about humans was removed.
L335-L340 Please remove the examples with chicks and horses and provide examples of how transport stress and weaning stress influenced both the microbiome and immune status in ruminants
(https://doi.org/10.3390/ani9090599, 10.3389/fmicb.2021.681014)
Based on the reviewer’s comment, the sentences about chicks and horses have been removed. Additionally, the following has been added and edited to fit into the paragraph, “Throughout livestock production, there are many stressful events (e.g., weaning and transportation) that are unavoidable. Research has shown after transportation, there is an increase in cortisol, adrenocorticotropic hormone (ACTH), and pro-inflammatory cytokines (i.e., IL-6, TNF-, IL-1) in multiple breeds of beef cattle (Li et al., 2019). This study also found abundances of ruminal Lactobacillus were positively correlated with IL-6 and IL-4. In addition, probiotics or prebiotics can be added to the diet of ruminants to prevent signaling of the HPA axis to increase anxiety. A study utilizing dairy calves found supplementing the calf’s diet with a multispecies probiotic prior to weaning improved growth, decreased incidence of diarrhea, affected the fecal microbiota (e.g., increased abundances of Bifidobacterium, Lactobacillus, Collinsella, and Saccharomyces), and reduced serum concentration (i.e., IgA, IgG, and IgM) (Wu et al., 2021).”
L341 Avoid informal language (so)
Based on the reviewer’s suggestion, “so” was replaced with “however.”
L352 negatively influence
Changed based on reviewer’s comment.
L356 the negative effects of rumen acidosis by (please describe the mode of action).
“… by increasing ruminal pH” was added to the end of the sentence based on the reviewer’s comment.
L358-L360 This statement is not valid, please remove it
This sentence was removed due to the reviewer’s suggestion.
L362-L364 in humans or animals? Please clarify
This sentence pertains to both since culturing methods are generally thought to be unable to culture the majority of commensal bacteria that comprise an environments microbiota. Since “individuals” seemed to suggest only humans, this has been changed to “host” to show it can be any mammal.
L376 what microbes, please use concise language
In order to clarify the microbes, this sentence has been reorganized to make it clearer. The new sentence is “The most abundant phylum of the LRT is Bacteroidetes and Firmicutes which is similar to the oral cavity microbiota suggesting the oral cavity plays a role in the development of the lung microbiota [131, 141-142].”
L395 This sentence needs to explain the connection between the microbiome and the respiratory tract and the GIT tract.
To further explain this relationship the phrase, “…by carrying bacteria and bacterial metabolites from the GIT to the lungs” was added at the end of this sentence.
L406 What microbes? Please provide examples
To further clarify the microbes being discussed here the following had been added to the sentence, “The commensal microbes comprising the microbiota of the respiratory tract…”
L419 What about ruminants? Organic acids like butyrate promote the growth and development of rumen papillae and prevent inflammation in the rumen. Please delete this sentence a provide examples with ruminants.
This sentence was replaced with the following to further emphasize the importance of butyrate in ruminants, “Butyrate supplementation has the ability to improve epithelial integrity while also in-crease the host’s defense mechanisms (Gailloteau et al., et al., 2010).”
L422-L423 This paragraph is unclear, please correct it
In order to improve the clarity of this paragraph it has been edited. It now says, “Due to the numerous stressors calves are exposed to during weaning, it serves as one of the most influential times for respiratory microbiota development [146-147, 168-169]. The composition of the URT is majorly affected by the first 40 days after arrival at the feedlot [170]. This is due to the stress, exposure to diseases, and dietary changes that occur when calves arrive at the feedlot that can result in dysbiosis in the URT which weakens a calf’s immune response and allow pathogens in the URT to migrate into the LRT [147, 155, 171-172].”
L426-L429 Please mention some of the bacterial or microbial communities observed across the respiratory tract and remove this paragraph. (See: 10.1038/srep29050)
Based on the reviewer’s suggestion, the following sentences have been added that summarize the suggested article, “In dairy calves experiencing illness, they had an increase in Mannheimia, Moraella, and Mycoplasma in their URT compared to that of healthy calves (Lima et al., 2016). Additionally, the URT of calves that later developed pneumonia was inhabited by a greater number of bacteria at 3 days of age than calves that remained healthy.”
L430-L439 This paragraph should explain the causes of BRD from the microbial standpoint providing examples of commensal microorganisms in the respiratory tract and their changes when the disease occurs. ( See https://doi.org/10.1186/s13567-021-01020-x)
An example of the bacteria associated with this disease was added t this paragraph. Also, an example using M. haemolytica was included to show how it damages the lungs. The additional sentences state, “One example of how these commensals cause disease is in the case of Mannheimia haemolytica. After the stressors occurring after arrival at a feedlot occur and cause dysbiosis, M. haemolytica rapidly proliferates within the URT (Chal et al., 2022). It then travels to the bronchial epithelial cells where it damages tight junction proteins, causes lesion in the lungs, releases leukotoxins and lipopolysaccharides which cause further damage to the respiratory tract, and triggers the host’s immune response causing inflammation.”
L441 This information is important from the physiological standpoint but with little relevance from the microbiome perspective. The information presented in this review should focus on the microbial aspects and physiological (https://doi.org/10.1186/s40168-020-00869-y)
Due to the reviewer’s comment about the viral information being relevant, the authors chose to remove this paragraph.
L499 There are better examples of mastitis with dairy ruminants, please correct this paragraph and provide a more detailed explanation. See 10.1038/s41598-017-08790-5 and https://doi.org/10.1371/journal.pone.0225001
The non-ruminant parts of this paragraph have been removed and the following has been added, “A study of healthy and infected dairy cows revealed the microbiota of milk from infected quarters has more variation compared to the milk microbiota of healthy quarter suggesting the milk microbiota experiences dysbiosis when infection occurs (Andrews et al., 2019). When a quarter is infected, the microbial community can become dominant by a specific microbial taxon. When comparing milk and skin microbiotas from cows with mastitis, Staphylococcus spp. and Debaryomyces spp. were shared between them (Andrews et al., 2019) indicating a link between the skin and milk microbiota.”
L515 Please revise the sentence and delete the parenthesis (check the whole manuscript)
This sentence was revised and now reads, “before the utilization of sequencing technologies, the reproductive tract of healthy females was considered sterile [192-194].”
L543-L546 Please avoid using examples with mice and non-ruminants, there are examples with ruminants (Please see https://doi.org/10.1016/j.vetmic.2022.109355)
The suggested sentences have been removed and the following one has replaced them, “Research in dairy cows found a genetic similarity in strains of E. coli found in the GIT and uterus suggesting the GIT aids in colonization of the uterine microbiota through ascending colonization of bacteria by the lower genital tract (Jones et al., 2022).”
L553-L572 Please explain the differences in the vagina microbiome between healthy ruminants compared to sick ruminants. (See https://doi.org/10.1016/j.rvsc.2021.11.007, https://doi.org/10.3168/jds.2019-17106, https://doi.org/10.1371/journal.pone.0200974 )
The suggested citations were utilized to add the additional sentences to this section, “Previous research in dairy cows found the uterine and vaginal microbiotas were most similar in composition 7 days postpartum (Miranda-CasoLuengo et al., 2019). Addition-ally, cows that later developed endometritis at 21 days postpartum had a delay in uterine and vaginal microbiota differentiation and a decrease in bacterial diversity at 7 days postpartum. The presence of Bacteriodes, Poryphomonas, and Fusobacteria were associated with metritis occurring after a decrease I n bacterial richness causing uterine dysbiosis (Galvao et al., 2019). This indicates these two environments may indicate future reproductive diseases. Specific bacterial families (Lachnospiraceae and Rikenellaceae) and genera (Acinetobacter, Bacillus, Oscillospira, CF231, and 5-7NS) have been identified as an indication of a healthy vaginal microbiota and have the potential to be a therapeutic target (Moreno et al., 2021).”
Round 2
Reviewer 1 Report
Figure 1 I think is on the verge of simplicity. In its present form, it does not offer anything to the reader. Or be removed or better enriched (indicating the main species of probiotics and, types of prebiotics, basic mechanisms that lead to a symbiotic state etc)
Author Response
Figure 1 I think is on the verge of simplicity. In its present form, it does not offer anything to the reader. Or be removed or better enriched (indicating the main species of probiotics and, types of prebiotics, basic mechanisms that lead to a symbiotic state etc)
Due to the reviewer’s comment about the simplicity of the figure, Figure 1 has been removed and Figure 2 has been renamed. The authors felt complicating the figure more would become to redundant with the text, so they felt it was best to just remove it.
Reviewer 2 Report
I revised the manuscript carefully; the authors improved on the previous version of the manuscript. However, the authors did not clarify some of the comments, and the explanations provided are incomplete. Please see the comments.
L44 Please revise the sentence, if there is a symbiotic relationship between the rumen microbes and the host it cannot be mainly mutualistic.
L85 The microbiota is healthy calves is not pathogenic, please revise and correct accordingly.
L88 Add a reference to support this statement
L109 The authors did not correct this statement, bacteria are not only classified as cellulolytic and amylolytic, but this is also incorrect. Please provide a more technical description of generalist and specialist microbes
L212-L214 This explanation is incomplete, please revise in detail the effects of subacute and acute rumen acidosis on rumen bacteria, the rumen wall, and the host.
L283 Add relevant information regarding the percentage of rumen microbes discovered by sequencing and the remaining proportion of unknown microbes.
L308-L320 The authors did not address the comment regarding removing the information related to humans and non-ruminants and provide the same examples but with ruminants. There are experiments conducted with ruminants on FMT.
L581 What sequencing methods? Please mention
L656-L665 Please correct the citation style
The authors have mentioned sequencing methods across the manuscript without providing examples, please provide a brief explanation somewhere after L581
Author Response
L44 Please revise the sentence, if there is a symbiotic relationship between the rumen microbes and the host it cannot be mainly mutualistic.
According to National Geographic, symbiotic relationships can be divided into 5 different types of relationships: mutualism, commensalism, predation, parasitism, and competition (https://education.nationalgeographic.org/resource/symbiosis-art-living-together). Due to this, symbiotic relationships can be mainly mutualistic in nature. Symbiosis as used in this context refers to the overarching umbrella term describing a relationship existing between the microbes in the rumen and host; whereas, the other types (in this case mutualism) describes the nature of the relationship.
L85 The microbiota is healthy calves is not pathogenic, please revise and correct accordingly.
When calves are very young and their microbiota is not yet developed, therefore, there is an opportunity for opportunistic pathogenic species to colonize the gastrointestinal tract. In order to clarify this statement, the word “opportunistic” has been included. However, the authors want to make a note that a newborn calf would not have what is considered a “healthy” functional rumen.
L88 Add a reference to support this statement
A reference supporting this has been included.
L109 The authors did not correct this statement, bacteria are not only classified as cellulolytic and amylolytic, but this is also incorrect. Please provide a more technical description of generalist and specialist microbes
Based on the reviewer’s comment, the sentence with the definitions has been expanded and now says, “Many bacterial species can be described as either generalist, bacteria able to degrade a wide variety of substrates and able to thrive in a wide variety of environments, or specialists, bacteria only able to degrade a very specific set of substrates that occupy a narrow niche [40].” Additionally, the sentence about cellulolytic and amylolytic was removed.
L212-L214 This explanation is incomplete, please revise in detail the effects of subacute and acute rumen acidosis on rumen bacteria, the rumen wall, and the host.
Based on the reviewer’s comment, the explanation about acidosis has been expanded. It now reads, “This digestive issue is caused when the rumen pH drops and becomes acidic as a result of an abundance of rapidly digestible carbohydrates in the diet producing an abundance of lactic acid (acute acidosis) or volatile fatty acids (subacute acidosis) [54]. As a result, the number of cellulolytic bacteria can decrease and the number of amylolytic bacteria can increase which can result in the microbiota experiencing dysbiosis [80]. Additionally, as the pH of the rumen decreases, the rumen epithelial cells become damaged preventing absorption of nutrients that can ultimately cause negative impact on the host’s performance and health [54].”
L283 Add relevant information regarding the percentage of rumen microbes discovered by sequencing and the remaining proportion of unknown microbes.
Depending on the sequencing technology utilized (e.g., 16S rRNA gene sequencing, whole genome sequencing) the percentage of microbes that are able to be identified within the gastrointestinal tract varies. Additionally, in order to develop gut therapies utilizing these microbial species, they must first be able to be cultured in a lab. Because of this the authors added the additional statement to the beginning of the next sentence in hopes to provide a better picture of the limitations of probiotics, “Currently we are only able to culture an estimated 23-40% of the microbes within the rumen [107]; therefore, we are limited in our ability to develop gut therapies.”
L308-L320 The authors did not address the comment regarding removing the information related to humans and non-ruminants and provide the same examples but with ruminants. There are experiments conducted with ruminants on FMT.
Due to the similarities between the FMT paragraph and the RFT paragraph, the authors chose to combine the two paragraphs about gut microbial transplants. They removed the examples about FMTs using humans and non-ruminants; however, the new examples using ruminants were moved and are no longer their own paragraph. They are now found between lines 697-700 and state, “Although not as commonly used in ruminants, FMT treatment can also be used in cattle. Research has shown a FMT can be an effective treatment for diarrhea [111]. Additionally, this study found that a FMT treatment in a calf’s early life can potentially improve growth performance.”
L581 What sequencing methods? Please mention
Due to the reviewer’s comment, the authors want to clarify what they mean when they mention sequencing technologies, so the addition of, “(e.g., whole genome sequencing, 16S rRNA gene sequencing)” was added to provide examples.
L656-L665 Please correct the citation style
The authors went back through and corrected any issues with citation style throughout the text.
The authors have mentioned sequencing methods across the manuscript without providing examples, please provide a brief explanation somewhere after L581
As stated above, the authors want to clarify what they mean when they mention sequencing technologies, so the addition of, “(e.g., whole genome sequencing, 16S rRNA gene sequencing)” was added to provide examples.